# FASTMTP: ACCELERATING LLM INFERENCE WITH ENHANCED MULTI-TOKEN PREDICTION

## ABSTRACT

As large language models (LLMs) become increasingly powerful, the sequential nature of autoregressive generation creates a fundamental throughput bottleneck that limits the practical deployment. While Multi-Token Prediction (MTP) has demonstrated remarkable benefits for model training efficiency and performance, its inherent potential for inference acceleration remains largely unexplored. This paper introduces FastMTP, a simple yet effective method that improves multi-step draft quality by aligning MTP training with its inference pattern, significantly enhancing speculative decoding performance. Our approach fine-tunes a single MTP head with position-shared weights on self-distilled data, enabling it to capture dependencies among consecutive future tokens and maintain high acceptance rates across multiple recursive draft steps. By integrating language-aware dynamic vocabulary compression into the MTP head, we further reduce computational overhead in the drafting process. Experimental results across seven diverse benchmarks demonstrate that FastMTP achieves an average of **2.03× speedup** compared to standard next token prediction with lossless output quality, outperforming vanilla MTP by 82%. FastMTP requires only lightweight training and seamlessly integrates with existing inference frameworks, offering a practical and rapidly deployable solution for accelerating LLM inference.

## 1 INTRODUCTION

Large Language Models (LLMs) have demonstrated remarkable capabilities (DeepSeek-AI et al., 2025; Yang et al., 2025a; GLM-4.5 Team et al., 2025; Team et al., 2025a) across diverse applications including autonomous agents (Wang et al., 2024; Guo et al., 2024), code generation (Liu et al., 2024b; Jiang et al., 2024), and complex reasoning tasks (Chen et al., 2025a; Ahn et al., 2024). However, their practical deployment faces a fundamental efficiency bottleneck: the autoregressive nature of token generation. Current LLMs generate text sequentially, producing only one token per forward pass, which means the overall generation time scales linearly with sequence length (Santilli et al., 2023). This becomes particularly problematic for scenarios requiring extensive generation, such as the state-of-the-art large reasoning models (OpenAI et al., 2024; DeepSeek-AI et al., 2025), which achieved breakthrough progress in solving complex and logic-intensive tasks by generating extended human-like Chain-of-Thoughts (CoTs) (Wei et al., 2022; Sprague et al., 2025) before reaching a final answer. While these models demonstrate strong reasoning capability, they often produce excessively long reasoning chains, even for simple samples, inevitably introducing substantial computational overhead (Feng et al., 2025). To this end, it highlights an urgent need for effective acceleration.

Recent research has explored multiple strategies to accelerate LLM inference, including efficient attention mechanisms (Katharopoulos et al., 2020; Child et al., 2019; Yang et al., 2024b; Lu et al., 2025; Dao, 2023) and model compression (Lin et al., 2024; Xiao et al., 2024; Gu et al., 2024; Hsieh et al., 2023) to reduce computational overhead. Some works have also focused on reducing CoT redundancy, including reinforcement learning with length penalties (Luo et al., 2025; Aggarwal & Welleck, 2025) and supervised fine-tuning on variable-length CoT data (Ma et al., 2025). Among these approaches, speculative decoding (Leviathan et al., 2023; Chen et al., 2023; Stern et al., 2018) has emerged as a promising technique that enhances decoding efficiency without compromising the fidelity of outputs. The core idea of this approach involves employing a smaller model, termed a

draft model, to predict several subsequent tokens that are then verified by the target LLM in parallel, achieving multi-token generation per forward pass (Zhou et al., 2024).

Multi-Token Prediction (MTP) (Gloeckle et al., 2024) modules and their auxiliary training, originally designed to improve training, offer a natural opportunity for inference acceleration through speculative decoding. MTP extends the traditional next-token prediction paradigm by training language models to predict multiple future tokens simultaneously. This approach encourages models to plan ahead and leverages richer supervision signals. Building on this foundation, DeepSeek-V3 (DeepSeek-AI et al., 2024) refined the MTP architecture with a sequential implementation using cascaded MTP modules, preserving the complete causal chain to maintain the autoregressive nature.

Despite growing adoption for training improvement (Team, 2025; GLM-4.5 Team et al., 2025; Team et al., 2025b; Xiaomi et al., 2025), the potential of MTP for inference acceleration remains largely unexploited. Current implementations either discard the MTP modules entirely during inference (DeepSeek-AI et al., 2024), reverting to standard next-token prediction, or keep only the first MTP module for multi-token prediction. This underutilization in inference may stem from two fundamental challenges in existing MTP implementations. First, the sequential MTP architecture requires cascaded forward passes through multiple MTP modules, with each module maintaining separate weights and key-value caches, resulting in memory overhead. Furthermore, while many inference frameworks now support speculative decoding with a single draft model, MTP's design necessitates loading and orchestrating multiple draft models—one MTP module for each prediction step—requiring complex scheduling that severely impacts computational efficiency. This may explain why models trained with multiple MTP layers often open-source only a single module for inference. Second, attempts to circumvent this complexity by recursively reusing a single MTP module yield poor acceptance rates beyond the first additional token, as this module was not explicitly trained for recursive multi-step prediction patterns, making it ineffective for extended draft generation.

In this paper, we present FastMTP, an enhanced multi-token prediction framework that makes the MTP module more effective, efficient, and deployable during inference. Our approach fine-tunes a single MTP head with shared weights across all prediction steps, teaching it to perform multi-token generation while maintaining causality. This enables the model to capture dependencies among consecutive future tokens, resulting in higher acceptance rates beyond the initial draft position, and ensures compatibility with EAGLE-style speculative decoding (Li et al., 2024a) for seamless integration with existing inference frameworks such as SGLang (Zheng et al., 2024). Inspired by FR-Spec (Zhao et al., 2025a), we integrate language-aware dynamic vocabulary compression that further reduces the computational cost of draft generation with negligible impact on acceptance rates across diverse tasks and languages.

Our key contributions are as follows:

- We adapt a single MTP head to perform effective recursive multi-step draft generation through fine-tuning on self-distilled data, dramatically improving average acceptance rates from 70% to 81% for the first draft token, from 11% to 56% for the second draft token and from 2% to 36% for the third token compared to the vanilla MTP reusing strategy, unlocking its full potential for inference acceleration.

- We adopt language-aware dynamic vocabulary compression for the MTP head, which reduces computational overhead during drafting according to the input context. It increases average output throughput by approximately 16% when drafting three additional tokens.

- We conduct extensive experiments on 7 benchmarks, demonstrating an average $2.03\times$ speedup on 7B models with lossless generation quality, significantly outperforming the vanilla MTP reusing strategy ($1.21\times$).

## 2 METHODOLOGY

In this section, we provide a detailed description of the implementation of FastMTP, which enhances MTP specifically for more effective and efficient speculative decoding during LLM inference. Figure 1 illustrates its training and inference pipelines.

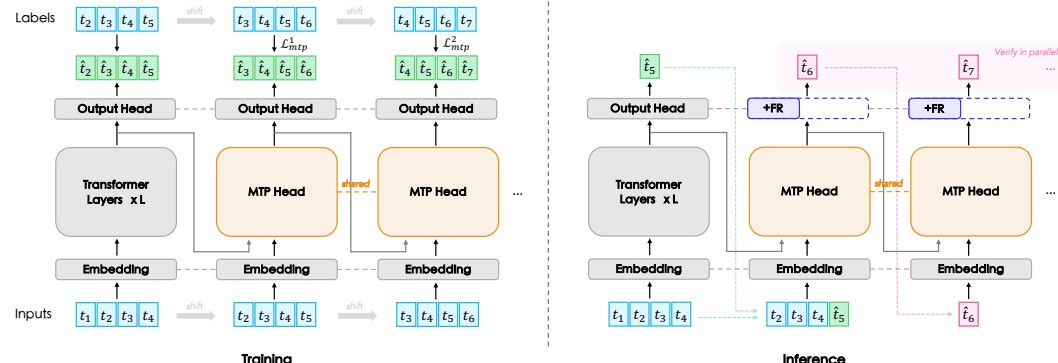

Figure 1: Illustration of FastMTP training and inference strategy. **Training phase** (left): Grey blocks are frozen main model modules, orange blocks are trainable MTP heads with shared weights, and blue blocks are input/label sequences with sequential shifts. **Inference phase** (right): The main model predicts the next token (green), which feeds into the MTP head for recursively generating draft tokens (pink) with parallel verification. Purple blocks indicate frequency-ranked (FR) language-aware dynamic vocabulary compression that accelerates draft generation.

## 2.1 TRAINING OF THE SHARED MTP HEAD

**Architecture Design.** FastMTP adopts the same MTP architecture as DeepSeek-V3 (DeepSeek-AI et al., 2024). The key distinction lies in our use of a single MTP head with shared weights across all prediction steps, instead of employing multiple independent modules for each prediction depth. This shared-weight design not only reduces memory usage, but more importantly, forces the model to capture dependencies among consecutive future tokens for causal multi-token prediction.

**Training Mechanism.** Let $\mathcal{F}$ denote the transformer layers of the main model and $\mathcal{M}$ denote the MTP head. Consider processing tokens at position $i$ in a sequence. The input tokens $t_{1:i}$ first pass through the embedding layer and transformer layers $\mathcal{F}$ to produce hidden states $h_{1:i}$. For predicting $K$ additional future tokens, the MTP head $\mathcal{M}$ operates recursively:

- At step $k = 1$: $\mathcal{M}$ takes the hidden state $h_i$ from $\mathcal{F}$ along with the embedding of the shifted token $t_{i+1}$ (shifted by 1 position) to predict token $\hat{t}_{i+2}^1$.

- At step $k > 1$: $\mathcal{M}$ processes its own output hidden state $h_{i+k-1}^{k-1}$ from step $k-1$, combined with the embedding of the shifted token $t_{i+k}$ (shifted by $k$ positions), to predict token $\hat{t}_{i+k+1}^k$.

During training, this process is applied to all valid positions in the sequence. For a training sequence of length $T$, position $i$ ranges from 1 to $T - K$ to ensure all indices remain valid after shifting.

**Training Objective.** We optimize the MTP head using a weighted cross-entropy loss with exponential decay for distant token predictions:

$$\mathcal{L}_{mtp} = \sum_{k=1}^{K} \alpha_k \cdot \mathcal{L}_{mtp}^k = \sum_{k=1}^{K} \alpha_k \cdot \text{CE}\left(\hat{t}_{1+k\,:\,T+1}^k,\ t_{1+k\,:\,T+1}\right) \tag{1}$$

where $\mathcal{L}_{mtp}^k$ represents the loss for the $k$-th prediction step, and $\text{CE}(\cdot, \cdot)$ denotes the cross-entropy loss between predicted and ground truth tokens. The position-dependent weights $\alpha_k$ follow an exponential decay:

$$\alpha_k = \frac{\beta^{k-1}}{\sum_{j=1}^{K} \beta^{j-1}} \tag{2}$$

where $\beta$ is the decay factor. This weighting strategy considers that the prediction uncertainty accumulates with each additional step – distant tokens depend on more intermediate decisions, making them progressively harder to predict. The exponential decay ensures the model prioritizes near-term predictions while still developing generation capabilities for several sequential future tokens.

Only the MTP head $\mathcal{M}$ is fine-tuned while all main model components remain frozen, ensuring both computational efficiency and preservation of the base model's capabilities.

## 2.2 LANGUAGE-AWARE VOCABULARY SPACE COMPRESSION IN MTP DRAFTING

To accelerate the drafting phase, we combine frequency-based vocabulary compression with MTP, reducing the computational overhead of the MTP output head. Following the analysis in FR-Spec (Zhao et al., 2025a), a small subset of tokens accounts for the vast majority of occurrences while the remaining tokens exhibit extremely sparse frequencies—a consistent long-tail pattern. This observation motivates us to restrict the MTP head's output space to high-frequency tokens during draft generation.

The original vocabulary compression methods primarily analyzed token distributions on English corpora. We find this severely limits performance on downstream non-English tasks. To overcome this, we compute language-specific frequency statistics and dynamically adjust high-frequency vocabularies based on context, ensuring adequate high-frequency tokens for different languages are represented in the compressed vocabulary space and thereby improving draft acceptance rates.

Let $\mathcal{V}$ denote the full vocabulary of the language model. For each language $l$, we define $\mathcal{V}_{high}^{(l)} \subset \mathcal{V}$ as the subset of high-frequency tokens specific to that language, identified through corpus-level statistics. In other words, if the main model's output head is $\mathbf{W} \in \mathbb{R}^{|\mathcal{V}| \times d}$, we extract the language-specific submatrix:

$$\tilde{\mathbf{W}}^{(l)}[i,:] = \mathbf{W}[\mathcal{V}_{high}^{(l)}[i],:], \quad i = 1,...,|\mathcal{V}_{high}^{(l)}| \tag{3}$$

During draft generation, the MTP head dynamically selects the appropriate vocabulary based on the input context, enabling focused compression that targets language-specific high-frequency patterns. With the extracted $\tilde{\mathbf{W}}^{(l)}$, the MTP head computes output logits only for tokens in $\mathcal{V}_{high}^{(l)}$ rather than the full vocabulary, reducing computational cost while maintaining high acceptance rates across different languages. Note that the vocabulary compression technique is restricted to the drafting phase, the verification phase retains the full vocabulary space, guaranteeing lossless generation quality.

## 2.3 EAGLE-STYLE MTP INFERENCE

The inference process differs from training by using autoregressively generated tokens from previous steps rather than teacher forcing. Given an input token sequence $t_{1:i}$, inference proceeds through two phases: draft generation and parallel verification.

**Drafting Phase.** In the draft generation phase, the input first passes through the main model to produce the next token $\hat{t}_{i+1}$, which serves as the first verified token. The MTP head $\mathcal{M}$ then autoregressively generates $K$ draft tokens following the EAGLE methodology (Li et al., 2024a):

- At the initial draft step $k = 1$: $\mathcal{M}$ takes the last hidden states from the transformer layers $\mathcal{F}$ along with the embedding of the concatenated sequence $[t_{2:i}; \hat{t}_{i+1}]$, where the newly predicted token is appended to the input sequence shifted one step ahead. This produces the first draft token $\hat{t}_{i+2}$.

- At the subsequent draft step $k > 1$: $\mathcal{M}$ operates autoregressively, processing its own output hidden state $\hat{h}_{i+k}$ from step $k-1$ combined with the embedding of the previously drafted token $\hat{t}_{i+k}$.

After $K$ recursive steps, we obtain the complete draft sequence $\hat{t}_{i+2:i+K+1}$.

**Verification Phase.** The main model then processes all draft tokens in parallel, computing logits for positions $i+2$ through $i+K+1$ simultaneously. Tokens are accepted sequentially until the first position where the draft token differs from what the main model would have sampled. Through strict verification without approximate strategies, we ensure that the output distribution remains identical to the original model, which has been theoretically proven (Leviathan et al., 2023; Chen et al., 2023), while achieving speedup through the acceptance of multiple tokens per forward pass.

## 2.4 TRAINING DATA

We employ a self-distillation approach for training data generation, where the main model itself generates all training responses. Inspired by prior works (Yang et al., 2024c; Cai et al., 2024), this strategy ensures natural alignment between the MTP head and the main model's distribution, leading to higher acceptance rates and more effective draft generation during speculative decoding.

Specifically, we collect diverse prompts from instruction-tuning datasets spanning multiple domains and languages. For each original sample $(x_n, y_n)$ where $x_n$ is the prompt and $y_n$ is the corresponding response from the source dataset, we generate a new response $\tilde{y}_n$ using the main model. The resulting self-distilled dataset $\{(x_n, \tilde{y}_n)\}$ captures the main model's semantic characteristics, generation patterns, and preferences. The detailed domain distribution, along with dataset curation and filtering procedures, is provided in Appendix A.

## 3 EXPERIMENTS

### 3.1 EXPERIMENTAL SETUP

**Model and Training Details.** FastMTP uses the pre-trained MiMo-7B-RL checkpoint (Xiaomi et al., 2025), a dense model with a single-layer MTP module and a 152K-token vocabulary (based on the Qwen2tokenizer (Yang et al., 2024a)). We freeze the main model (including the transformer layers, embedding layer, and output head), fine-tuning only the MTP head's 210.8M parameters, which accounts for less than 3% of the backbone model's total parameters. The MTP head was trained for 3 epochs on 389.4K self-distilled samples (see Appendix A for dataset details). Training employed cosine learning rate scheduling with a peak learning rate of 5e-5, a warmup ratio of 0.05, and a global batch size of 64. For the MTP-specific hyperparameters, we used a loss weight decay factor $\beta = 0.6$ and prediction depth $K = 3$. Training was conducted using the ms-swift framework (Zhao et al., 2025b). The entire training process was completed in less than 1 day on a single H20 server, demonstrating low training cost.

**Evaluation Methods.** We evaluate FastMTP on seven tasks adapted from Spec-Bench (Xia et al., 2024): MT-Bench (Zheng et al., 2023) for multi-turn conversation, LiveCodeBench-v6 (Jain et al., 2024) for coding, MATH-500 (Lightman et al., 2023) for mathematical reasoning, Natural Questions (Kwiatkowski et al., 2019) for both RAG and question answering, CNN/Daily Mail (Nallapati et al., 2016) for summarization, and C-Eval (Huang et al., 2023) for Chinese knowledge assessment. Details of these evaluation benchmarks are provided in Appendix B. All evaluations employ strict speculative decoding acceptance criteria, ensuring no degradation in generation quality. All experiments were conducted using the SGLang inference framework (Zheng et al., 2024) with single-batch inference, greedy decoding (temperature 0), and a maximum generation length of 1024 tokens across all tasks. We conducted our primary experiments on an NVIDIA A10 GPU (24GB), a widely-deployed accelerator in production environments.

**Metrics.** We use four widely used metrics for evaluation: (1) Average acceptance length $\tau$: mean accepted tokens per forward pass of the main model. (2) Acceptance Rate: the percentage of draft tokens accepted during verification. (3) Average output throughput: mean output tokens per second (token/s). (4) Speedup ratio: relative speedup compared to baseline (vanilla autoregressive decoding). Note that speculative decoding theoretically preserves the main model's output distribution (Leviathan et al., 2023; Chen et al., 2023). Since we employ strict acceptance criteria without any relaxation, the outputs are identical to vanilla autoregressive decoding, making separate generation quality evaluation unnecessary.

Table 1: Average Accepted Length and Decoding Speed (token/s) for MiMo-RL-7B Using Various Methods to Predict K Draft Tokens. Tasks include multi-turn conversation (MT.), coding (Code), mathematical reasoning (Math), retrieval-augmented generation (RAG), question answering (QA), summarization (Summ.), and Chinese knowledge (ZH.)

| | Method | MT. | | Code | | Math | | RAG | | QA | | Summ. | | ZH. | | Mean | |
|---|---|---|---|---|---|---|---|---|---|---|---|---|---|---|---|---|---|
| | | $\tau$ | token/s | $\tau$ | token/s | $\tau$ | token/s | $\tau$ | token/s | $\tau$ | token/s | $\tau$ | token/s | $\tau$ | token/s | $\tau$ | token/s |
| K=0 | Baseline | 1.00 | 31.28 | 1.00 | 31.49 | 1.00 | 31.90 | 1.00 | 31.27 | 1.00 | 31.92 | 1.00 | 31.42 | 1.00 | 31.63 | 1.00 | 31.55 (1.00x) |
| K=1 | Vanilla MTP | 1.72 | 41.86 | 1.73 | 42.71 | 1.83 | 44.41 | 1.65 | 40.22 | 1.67 | 41.06 | 1.64 | 39.78 | 1.67 | 41.66 | 1.70 | 41.67 (1.32x) |
| | Fixed-data FT | 1.75 | 43.46 | 1.80 | 44.67 | 1.86 | 45.06 | 1.78 | 43.17 | 1.71 | 41.85 | 1.72 | 42.40 | 1.74 | 42.70 | 1.76 | 43.33 (1.37x) |
| | Self-data FT | 1.80 | 44.51 | 1.84 | 45.39 | 1.90 | 46.43 | 1.82 | 43.99 | 1.74 | 42.67 | 1.77 | 42.97 | 1.76 | 44.30 | 1.81 | 44.32 (1.40x) |
| | Self-data FT + FR | 1.78 | 45.79 | 1.82 | 47.59 | 1.88 | 47.58 | 1.80 | 44.60 | 1.74 | 44.60 | 1.74 | 45.45 | 1.76 | 46.38 | 1.79 | 46.15 (1.46x) |
| K=2 | Vanilla MTP | 1.85 | 42.03 | 1.82 | 41.70 | 2.00 | 45.52 | 1.71 | 38.35 | 1.82 | 41.26 | 1.73 | 39.00 | 1.78 | 41.27 | 1.81 | 41.30 (1.31x) |
| | Fixed-data FT | 2.20 | 49.83 | 2.33 | 53.34 | 2.52 | 56.51 | 2.31 | 51.65 | 2.10 | 47.47 | 2.14 | 47.29 | 2.19 | 49.85 | 2.26 | 50.85 (1.61x) |
| | Self-data FT | 2.35 | 53.90 | 2.45 | 55.67 | 2.62 | 59.68 | 2.40 | 53.96 | 2.23 | 50.71 | 2.26 | 50.93 | 2.27 | 52.66 | 2.37 | 53.93 (1.71x) |
| | Self-data FT + FR | 2.30 | 58.12 | 2.40 | 60.44 | 2.59 | 63.71 | 2.37 | 58.63 | 2.21 | 55.47 | 2.19 | 55.02 | 2.24 | 58.20 | 2.33 | 58.51 (1.85x) |
| K=3 | Vanilla MTP | 1.86 | 38.97 | 1.83 | 38.17 | 2.02 | 42.06 | 1.72 | 35.27 | 1.83 | 38.10 | 1.75 | 36.17 | 1.78 | 37.58 | 1.83 | 38.04 (1.21x) |
| | Fixed-data FT | 2.48 | 51.53 | 2.64 | 55.40 | 2.93 | 59.72 | 2.62 | 53.97 | 2.30 | 48.08 | 2.35 | 48.38 | 2.45 | 51.69 | 2.54 | 52.68 (1.67x) |
| | Self-data FT | **2.69** | 56.33 | **2.85** | 59.49 | **3.16** | 65.94 | **2.80** | 57.55 | **2.49** | 52.31 | **2.55** | 53.08 | **2.55** | 54.36 | **2.73** | 57.01 (1.81x) |
| | Self-data FT + FR | 2.62 | **63.42** | 2.75 | **66.24** | 3.07 | **73.66** | 2.75 | **64.55** | 2.46 | **58.88** | 2.47 | **59.15** | 2.53 | **62.93** | 2.66 | **64.12 (2.03x)** |

## 3.2 MAIN RESULTS

We evaluate three primary configurations: (1) vanilla autoregressive decoding *(baseline)*, (2) the original MTP checkpoint without fine-tuning *(vanilla MTP)*, and (3) the proposed FastMTP. To analyze the contribution of each component, we further evaluate FastMTP variants:

- **Fixed-data FT**: Fine-tunes using both prompts and responses from original data sources.
- **Self-data FT**: Employs self-distilled responses generated by the main model.
- **Self-data FT + FR**: Additionally incorporates language-aware vocabulary compression.

We assess performance across different draft depths $K \in \{0, 1, 2, 3\}$, where $K = 0$ represents vanilla autoregressive decoding without drafting, and $K = 3$ indicates three recursive MTP forward passes generating three additional draft tokens.

Table 1 and Figure 2 present comprehensive acceleration performance evaluations of FastMTP across seven diverse tasks. FastMTP (Self-data FT + FR) achieves superior performance across all benchmarks, delivering an average 2.03× speedup over *baseline* at $K = 3$. The speedup gains vary across task domains, reflecting their distinct generation characteristics. Mathematical reasoning exhibits the highest speedup and average acceptance length ($\tau = 3.16$ before vocabulary compression), demonstrating the MTP head's effectiveness in capturing structured reasoning patterns. Coding tasks deliver the second-best performance, benefiting from prevalent fixed templates and repetitive programming constructs. General NLP tasks such as question answering show comparatively smaller improvements (1.84×–2.07×), potentially due to their higher linguistic diversity and less predictable token dependencies. This task-specific variation, visualized in Figure 2, confirms that FastMTP maintains consistent acceleration across diverse generation scenarios.

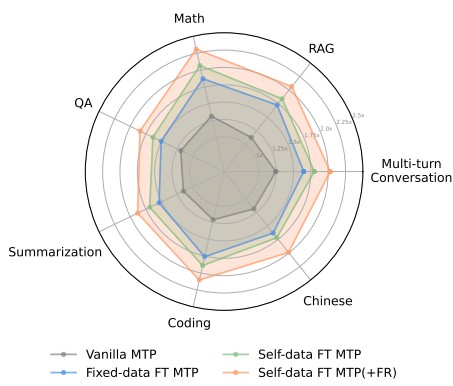

Figure 2: Speedup comparison of various methods across subtasks, evaluated on a single A10 GPU.

## 3.3 FURTHER ANALYSES

Our further analyses explore two critical aspects of FastMTP's design and performance. (1) Optimal draft length: What is the optimal number of MTP draft steps for achieving maximum speedup? In other words, what is the effective prediction distance that a single MTP head can learn while main-

taining high acceptance rates? (2) Vocabulary compression trade-offs: What degree of vocabulary compression achieves the optimal balance between computational efficiency and acceptance rate? How does this trade-off vary across different languages?

### 3.3.1 OPTIMAL DRAFT LENGTH

To determine the optimal number of draft steps, we trained an MTP head capable of predicting up to 7 additional tokens. We evaluated the decoding speed and acceptance length for different drafting length $K$. As shown in Figure 3, FastMTP achieves peak speedup at $K = 3$, reaching 140 token/s, while the vanilla MTP checkpoint peaks earlier at $K = 2$ with lower output throughput. For the acceptance length, FastMTP shows consistently growing acceptance length from 1.0 to 3.2 as $K$ increases, demonstrating that our training strategy successfully enables the model to learn multi-step predictions. In contrast, vanilla MTP maintains a nearly flat acceptance length around 1.8 from $K = 2$, indicating poor prediction quality beyond the first position.

Despite the monotonically increasing acceptance length, the decoding speed peaks at $K = 3$ and then gradually declines. While longer drafts increase acceptance length, they also incur additional computational overhead. Beyond $K = 3$, the marginal gains in acceptance length cannot compensate for the growing computational overhead of generating longer drafts, as distant tokens become progressively harder to predict accurately, which reduces overall efficiency. Therefore, **FastMTP achieves maximum speedup at draft length K = 3, where the acceptance length improvements through training optimally outweigh the computational overhead of recursive drafting.**

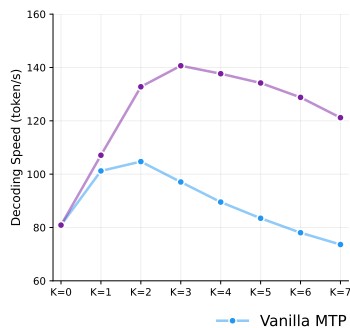 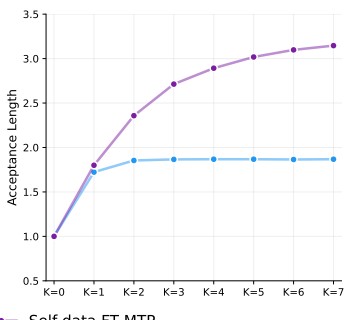

Vanilla MTP    Self-data FT MTP

Figure 3: Decoding Speed and Acceptance Length with different MTP draft length on a single A100.

### 3.3.2 VOCABULARY COMPRESSION TRADE-OFFS

To investigate the impact of vocabulary compression, we evaluated four vocabulary sizes compared to the full vocabulary. We select two representative benchmarks for language-specific analysis: C-Eval (Huang et al., 2023) for Chinese knowledge question answering and MT-Bench (Zheng et al., 2023) for English multi-turn conversation.

Vocabulary frequency patterns are observed to be highly language-dependent. Original implementations using token statistics from the English-dominated corpus SlimPajama-627B (Daria et al., 2023) yield low acceptance rates and consequently poor performance on real-world Chinese tasks due to mismatched frequency distributions, confirming the context-related acceleration paradigm (Zhao et al., 2025a). To address

Table 2: Different FR-Spec configurations.

|  |  | $\tau$ | token/s | speedup |
|---|---|---|---|---|
| C-Eval | Baseline | 1.000 | 31.627 | 1.000x |
|  | Full Vocab. (152k) | 2.551 | 54.364 | 1.719x |
|  | +FR 8k | 2.448 | 62.015 | 1.961x |
|  | **+FR 16k** | **2.525** | **62.928** | **1.990x** |
|  | +FR 32k | 2.549 | 62.210 | 1.967x |
|  | +FR 64k | 2.550 | 60.083 | 1.900x |
| MT-Bench | Baseline | 1.000 | 31.276 | 1.000x |
|  | Full Vocab. (152k) | 2.690 | 56.333 | 1.801x |
|  | +FR 8k | 2.426 | 60.260 | 1.927x |
|  | +FR 16k | 2.519 | 62.144 | 1.987x |
|  | **+FR 32k** | **2.622** | **63.420** | **2.028x** |
|  | +FR 64k | 2.677 | 61.572 | 1.969x |

this, we build language-specific compressed vocabularies. For Chinese tasks, we select the Chinese-DeepSeek-R1-Distill-110k-SFT dataset (Liu et al., 2025), which contains high-quality and diverse Chinese instruction-following samples, to identify high-frequency tokens in Chinese. During infer-

ence, FastMTP dynamically switches to appropriate compressed vocabularies according to the input context.

Table 2 presents the performance across different vocabulary configurations. We observe that even aggressive vocabulary compression incurs only minimal degradation in acceptance length while delivering considerable speedup gains. Notably, the optimal compression ratio varies across languages. For MT-Bench, 32k vocabulary achieves peak performance with $2.028\times$ speedup, sacrificing merely 0.068 tokens in acceptance length compared to the full vocabulary. In contrast, C-Eval achieves optimal performance with a more aggressive 16k vocabulary, yielding $1.990\times$ speedup with an even smaller acceptance length reduction of 0.026 tokens. This suggests that Chinese may exhibit more concentrated token usage patterns in large vocabulary, making it benefit more substantially from vocabulary compression strategies. In conclusion, **the optimal vocabulary size differs by language—16k for Chinese, 32k for English—revealing distinct token distribution patterns.**

### 3.4 ABLATION EXPERIMENTS

We conduct comprehensive ablation experiments to validate the contribution of each component in FastMTP and analyze the acceptance rates of each draft position after training.

**Impact of Design Choices.** To quantify the individual contributions of our design choices, we evaluate three configurations (detailed in Section 3.2) against the vanilla MTP. As shown in Table 1, FastMTP achieves a $2.03\times$ speedup, representing an 82% improvement over vanilla MTP ($1.21\times$), a 36% gain over fixed-data fine-tuning ($1.67\times$), and a 22% increase over self-distilled data fine-tuning without vocabulary compression ($1.81\times$). These incremental improvements validate the effectiveness of each component: fine-tuning enables more accurate multi-token drafting during inference, self-distillation achieves better distribution alignment between MTP head and main model for higher acceptance length, and vocabulary compression reduces computational overhead to increase output throughput with minimal impact on acceptance rates.

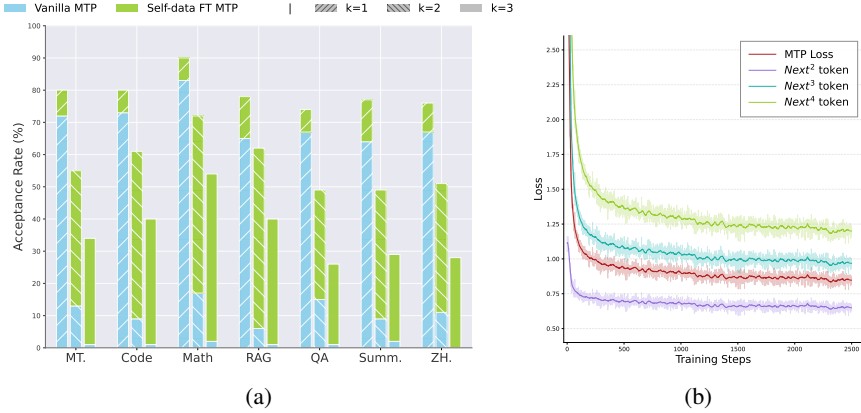

(a)                  (b)

Figure 4: (a) Acceptance rate improvement of self-distilled dataset fine-tuned MTP compared with vanilla MTP at different draft steps. (b) Training Loss of MTP at different draft steps.

**Acceptance Rate Analysis.** Figure 4a further demonstrates the critical role of fine-tuning in enabling effective multi-token drafting. Vanilla MTP exhibits severe performance degradation beyond the first draft step, with acceptance rates dropping sharply from approximately 70% at $k = 1$ to merely 10% at $k = 2$, and approaching zero at $k = 3$. This collapse exposes the inherent limitation of deploying vanilla MTP checkpoints for recursive multi-token prediction—it was trained only for single-step ahead prediction and thus fails to maintain high acceptance rates at deeper draft positions. In contrast, our fine-tuned MTP achieves substantially higher acceptance rates across all draft steps: 80% at $k = 1$, 56% at $k = 2$, and 36% at $k = 3$ on average. Mathematical reasoning shows the highest gains, maintaining over 50% acceptance rate even at $k = 3$ compared to vanilla MTP's mere 3%. The consistent superiority across all seven tasks validates that fine-tuning enables the MTP head to learn dependencies among consecutive future tokens for accurate multi-step pre-

diction. Figure 4b illustrates the training loss dropping sharply in the early stage, followed by slower but steady decrease until convergence. As expected, deeper positions showing higher losses, further illustrating that distant tokens become progressively harder to predict accurately.

# 4 RELATED WORK

## 4.1 LLM INFERENCE ACCELERATION

The growing computational demands of LLMs have motivated extensive research into inference acceleration techniques, which can be broadly categorized into four main directions: architectural innovations, model compression, framework optimizations, and speculative decoding. Architectural innovations focus on redesigning model components for enhanced efficiency. Representative works include efficient attention mechanisms that reduce quadratic complexity, such as linear attention (Katharopoulos et al., 2020; Yang et al., 2025b), sparse attention (Child et al., 2019; Lu et al., 2025), and low-rank attention (DeepSeek-AI, 2024). Model compression techniques reduce computational and memory requirements, including quantization (Frantar et al., 2023; Lin et al., 2024; Dettmers et al., 2022; Xiao et al., 2024), pruning (Frantar & Alistarh, 2023; Ma et al., 2023), and knowledge distillation (Gu et al., 2024; Hsieh et al., 2023). Framework optimizations improve deployment efficiency through system-level engineering. Notable contributions include fused operators (Dao et al., 2022), PagedAttention (Kwon et al., 2023),RadixAttention (Zheng et al., 2024), and various parallelism strategies (NVIDIA, 2023). Speculative decoding (Leviathan et al., 2023; Chen et al., 2023; Stern et al., 2018) leverages draft-then-verification paradigms to accelerate decoding.

## 4.2 SPECULATIVE DECODING OPTIMIZATION

Optimization efforts in speculative decoding primarily target two aspects: obtaining higher acceptance rates, and reducing draft generation overhead.

To achieve higher acceptance rates, prior research has explored two primary directions. The first focuses on improving draft quality through specialized model design and training. Several works adopt self-drafting by adding additional prediction heads to the target model. For example, Medusa (Cai et al., 2024) employs extra MLP heads that reuse the last hidden states from the target model to predict the next few tokens in parallel, while EAGLE (Li et al., 2024a;b; 2025)incorporates both hidden state and the preceding tokens to draft in an autoregressive way, significantly improving the stability and accuracy. Other works utilize smaller models from the same model series as draft model (Leviathan et al., 2023; Chen et al., 2025b), exploiting similarity for better alignment. The second direction slightly relaxes the matching requirement, leading to higher acceptance of drafted tokens (Xia et al., 2024). For instance, SpecDec (Xia et al., 2023) only requires the drafted tokens to fall in top-$\beta$ candidates with a tolerable score gap away from the top-1 result.

To reduce draft generation overhead, one line of work focuses on dynamic draft tree construction. BiLD (Kim et al., 2023) and Kangaroo (Liu et al., 2024a) implement early stopping to control tree depth, while EAGLE-2 (Li et al., 2024b) leverage confidence scores to approximate acceptance rates and adjust draft tree structures. Another line of work has targeted computational bottlenecks in the drafting process itself. For example, TriForce (Sun et al., 2024) accelerates drafting through KV-cache compression, Ouroboros (Zhao et al., 2024) adapts lookahead decoding (Fu et al., 2024) techniques, and FR-Spec (Zhao et al., 2025a) identifies the computational bottleneck of large-vocabulary LM heads, restricting the drafting space to high-frequency token subsets.

# 5 CONCLUSION

In this work, we propose FastMTP as an improvement over vanilla multi-token prediction in the speculative decoding during LLMs inference. FastMTP introduces two key implementations. First, we train a single shared-weight MTP head through self-distillation, eliminating the need for multiple independent MTP modules while improving multi-step prediction capability. Second, we employ language-aware vocabulary compression to reduce computational costs during draft generation. Experimental results demonstrate the superiority of the proposed method, where both draft token quality and overall output speedup can be enhanced in a series of scenarios.

REPRODUCIBILITY STATEMENT

To ensure full reproducibility, we release our source code, model checkpoints, language-specific compressed vocabularies, and experimental configurations. Our repository includes detailed instructions for environment setup and code execution, with thorough checking. Experimental details—including model configurations, training procedures, evaluation methods, metrics, and loss curves—are described in Section 3. Dataset generation, processing, and mixing procedures, along with evaluation benchmarks, are documented in Appendix A and B. All resources are publicly available to enable rapid deployment, result replication, and future extensions of our work.

ETHICS STATEMENT

Our research involves no human subjects and relies only on publicly available models and datasets in compliance with their original licenses and terms of use. We recognize that these backbone models and datasets may contain inherent biases, which our acceleration method does not address or mitigate. We acknowledge that our work could be applied to various downstream applications and encourage responsible deployment.

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

## A    TRAINING DATASETS DETAILS

We collect our dataset for MTP head fine-tuning from a variety of data sources. To ensure distribution alignment, we employ self-distillation where all responses are generated by the main model itself, ensuring the MTP head learns to produce draft tokens consistent with the main model's behavior, rather than attempting to mimic responses from external sources that may have different distributions. For each prompt $x_n$ extracted from the source datasets, we generate corresponding response $\tilde{y}_n$ using the main model with the following generation configurations: temperature of 0.6, top-k of 20, top-p of 0.95, and maximum length of 4096 tokens. The entire data distillation process is conducted using the SGLang inference framework (Zheng et al., 2024).

After distillation, we apply de-duplication, data cleaning, and mixing strategies to curate high-quality tokens:

- **De-duplication:** We perform global MinHash de-duplication both within individual data sources and across the entire dataset to remove near-duplicate samples.
- **Data cleaning:** We develop heuristics to filter out low-quality samples. Some examples of heuristics include: (1) samples with incomplete or truncated reasoning chains; (2) samples containing excessive repetitive content; and (3) samples that fall outside desired length ranges.
- **Data mixing:** Our final dataset comprises four major categories with carefully balanced proportions: approximately 42% of tokens corresponding to general knowledge and tasks, 18% to mathematical and reasoning content, 13% to code, and 27% to Chinese texts, yielding the final 389.4K high-quality training examples.

## B    EVALUATION BENCHMARKS DETAILS

We evaluate FastMTP across seven diverse tasks to ensure comprehensive coverage of real-world applications: multi-turn conversation, code generation, mathematical reasoning, retrieval-augmented generation (RAG), question answering, summarization, and Chinese knowledge assessment. Adapting from the Spec-Bench evaluation framework (Xia et al., 2024), we carefully select benchmarks that represent distinct generation patterns and challenges. To ensure fair evaluation, we randomly sample 80 instances from each task (102 for C-Eval) and avoid benchmarks potentially exposed during MTP training.

Our benchmark suite comprises:

- **MT-Bench** (Zheng et al., 2023): A series of open-ended questions that evaluate a chatbot's multi-turn conversational and instruction-following ability. MT-bench is also carefully constructed to differentiate chatbots based on their core capabilities, such as reasoning and math.
- **LiveCodeBench-v6** (Jain et al., 2024): A comprehensive and contamination-free evaluation of LLMs for code collects new problems over time from contests across three competition platforms, namely LeetCode, AtCoder, and CodeForces. We select the sixth version of the dataset, which contains 1055 problems released between May 2023 and Apr 2025.
- **MATH-500** (Lightman et al., 2023): A dataset comprising 500 challenging mathematical problems designed to test advanced mathematical reasoning and problem-solving skills. It includes problems from various domains such as algebra, calculus, geometry, and number theory, primarily at high school and early undergraduate levels.
- **Natural Questions** (Kwiatkowski et al., 2019): A large-scale QA dataset containing real user queries paired with high-quality annotations from Wikipedia documents. We utilize this benchmark in two subtasks: question answering and retrieval-augmented generation (RAG).
- **CNN/Daily Mail** (Nallapati et al., 2016): An English-language dataset containing just over 300k unique news articles as written by journalists at CNN and the Daily Mail, each with 3-4 highlights that summarize the contents of the article. This benchmark evaluates the model's ability to produce extractive and abstractive summarizations of news articles while preserving key information.

- **C-Eval** (Huang et al., 2023): A comprehensive Chinese evaluation suite designed to assess advanced knowledge and reasoning abilities of foundation models in a Chinese context. C-Eval comprises multiple-choice questions across four difficulty levels: middle school, high school, college, and professional. The questions span 52 diverse disciplines, ranging from humanities to science and engineering.

## C  USE OF LLMS

We used large language models for minor text polishing and grammatical error correction during manuscript preparation. All scientific content, experimental work, analysis, and intellectual contributions are entirely the authors' original work.

