# OpenReview forum: "FastMTP: Accelerating LLM Inference with Enhanced Multi-Token Prediction"
_ICLR.cc/2026/Conference — ICLR 2026 Conference Withdrawn Submission_

### Official Review · Reviewer_Xj93 · 2025-10-14

**Soundness:** 2
**Presentation:** 3
**Contribution:** 2
**Rating:** 2
**Confidence:** 5

**Summary:**

The paper introduces FastMTP, a method that refines MTP for inference by training a single shared MTP head on self-distilled data, so it better captures dependencies among consecutive future tokens and boosts speculative decoding performance. To reduce computational overhead, FastMTP also applies language-aware dynamic vocabulary compression in the draft stage. Evaluated across seven benchmarks, it achieves ~2.0× average speedup over standard next-token prediction without loss in output quality.

**Strengths:**

The paper is clearly written with good-quality diagrams like Figure 1 and methodology is well written. Speedup is achieved over vanilla MTP.

**Weaknesses:**

1. My main concern with this paper is its novelty. The proposed method does not seem to be very different from Medusa (https://arxiv.org/abs/2401.10774) or Hydra (https://arxiv.org/abs/2402.05109). Could you probably explain the major differences?
2. There is a lack of discussion of relevant work on parallel decoding. For example, BiTA (https://arxiv.org/abs/2401.12522) and PPD (https://arxiv.org/abs/2405.18628) use prompt tokens for MTP. What are the advantages of FastMTP as compared to these existing methods?
3. The speedup ratio is not compared with SOTA speculative decoding methods like Eagle (https://arxiv.org/abs/2503.01840). What is the motivation of FastMTP if the speedup ratio seems far away from Eagle?

**Questions:**

Please see the weaknesses mentioned above.

---

### Official Review · Reviewer_Xx1p · 2025-10-31

**Soundness:** 2
**Presentation:** 3
**Contribution:** 2
**Rating:** 2
**Confidence:** 4

**Summary:**

This paper introduces FastMTP, a simple yet effective method that improves MTP performance by aligning MTP training with its inference. It fine-tunes the original MTP head of MiMo-7B-RL on 389.4K self-distilled samples.  Experimental results across seven diverse benchmarks demonstrate that FastMTP achieves an average of 2.03x speedup compared to standard next token prediction with lossless output quality, outperforming vanilla MTP head by 82%.

**Strengths:**

1. The exploration to improve existing MTP performence in practice should be encouraged.
2. The improvement is promising, which shows that FastMTP achieves an average of 2.03x speedup compared to standard next token prediction with lossless output quality, outperforming vanilla MTP by 82%.
3. The manuscript is clearly written, with a well-structured narrative.

**Weaknesses:**

1. **Lack of technical contributions**: The main contribution of this work appears to be fine-tuning the original MTP head of MiMo-7B-RL on 389.4K self-distilled samples. The model architecture, training objective, and inference process are similar to the EAGLE series [1]. If that is the case, the methodology should primarily highlight the construction of the self-distilled dataset. However, this version of the manuscript devotes large parts of the main body to presenting content similar to existing speculative approaches, which lack substantial technical contributions.
2. **Stronger baselines**: FastMTP fine-tunes the original MTP head of MiMo-7B-RL on 389.4K self-distilled samples. The training process is identical to the conventional training recipe for speculative decoding (SD). Given this, FastMTP should be compared to stronger SD baselines, such as the EAGLE series, to better demonstrate its superiority and contributions.
3. **Lack of data construction details**: The main contribution of this manuscript is the construction of 389.4K self-distilled samples. However, this process is only briefly described in Appendix A, with many important details missing. For example:
   - What exact data sources are used to construct the dataset? These sources should be clearly introduced and properly cited.
   - How are the proportions of the different data sources determined?
   - What is the total token count of the 389.4K self-distilled samples?



[1] EAGLE: Speculative Sampling Requires Rethinking Feature Uncertainty. Li et al. ICML 2024.

**Questions:**

Please check the weakness part above.

---

### Official Review · Reviewer_KsDF · 2025-11-01

**Soundness:** 3
**Presentation:** 3
**Contribution:** 2
**Rating:** 4
**Confidence:** 4

**Summary:**

The paper proposes the use of self-distillation to train a single MTP to predict several future drafts. The authors also incorporate vocabulary compression to improve the acceptance rate of the drafts. FastMTP outperforms vanilla MTP by reducing inference latency and improving acceptance rate.

**Strengths:**

* The algorithm makes use of the MTP modules obtained during pretraining which are usually aligned with the primary model’s representations and thus increase the acceptance rate.
* The paper reduces the memory requirements over vanilla MTP by having a single module for predicting several draft tokens.
* The algorithm improves over vanilla MTP’s acceptance rate and reduces latency.

**Weaknesses:**

* The algorithm does not scale beyond K=3 draft tokens. In real world deployment scenarios with a single primary model and multiple drafters, going beyond K=3 may be required.
* The paper does not compare with other speculative decoding methods Sequoia, SpecDec++, Eagle-3, Medusa, QuantSpec, especially with algorithms such as QuantSpec that use self-speculation.

References
* Sequoia: Scalable, Robust, and Hardware-aware Speculative Decoding, Chen et al., 2024.
* EAGLE-2: Faster Inference of Language Models with Dynamic Draft Trees, Le et al., 2024.
* SpecDec++: Boosting Speculative Decoding via Adaptive Candidate Lengths, Huang et al.,  2024.
* Medusa: Simple LLM Inference Acceleration Framework with Multiple Decoding Heads, Cai et al., 2024.
* QuantSpec: Self-Speculative Decoding with Hierarchical Quantized KV Cache, Tiwari et al., 2025.

**Questions:**

* Does training for K > 3 improve the acceptance rate (even for K<=3)?
* Is there an ablation on the number of MTP modules? Do 2 MTP modules trained with self distillation offer a more prominent reduction in latency when compared to 1?
* Is there an ablation around initializing the MTP module from scratch? For models not trained with MTP, this ablation can help assess the universality of FastMTP.

---

### Note · Authors · 2025-11-25

I have read and agree with the venue's withdrawal policy on behalf of myself and my co-authors.